# Genomic Surveillance of Clinical *Pseudomonas aeruginosa* Isolates Reveals an Additive Effect of Carbapenemase Production on Carbapenem Resistance

Luke Diorio-Toth,[a] Sidra Irum,[b] Robert F. Potter,[a,c] Meghan A. Wallace,[c] Muhammad Arslan,[d] Tehmina Munir,[e] Saadia Andleeb,[b] Carey-Ann D. Burnham,[c,f,g,h] Gautam Dantas[a,c,g,i]

aThe Edison Family Center for Genome Sciences and Systems Biology, Washington University School of Medicine, St. Louis, Missouri, USA

bAtta ur Rahman School of Applied Biosciences, National University of Sciences and Technology, Islamabad, Pakistan

cDepartment of Pathology and Immunology, Washington University School of Medicine, St. Louis, Missouri, USA

dPakistan Institute of Medical Sciences (PIMS), Islamabad, Pakistan

eDepartment of Microbiology, Army Medical College, National University of Medical Sciences, Rawalpindi, Pakistan

fDepartment of Medicine, Washington University School of Medicine, St. Louis, Missouri, USA

gDepartment of Molecular Microbiology, Washington University School of Medicine, St. Louis, Missouri, USA

hDepartment of Pediatrics, Washington University School of Medicine, St. Louis, Missouri, USA

iDepartment of Biomedical Engineering, Washington University in St. Louis, St. Louis, Missouri, USA

Luke Diorio-Toth and Sidra Irum contributed equally to this article. Author order was determined alphabetically.

**ABSTRACT** Carbapenem resistance in *Pseudomonas aeruginosa* is increasing globally, and surveillance to define the mechanisms of such resistance in low- and middle-income countries is limited. This study establishes the genotypic mechanisms of $\beta$-lactam resistance by whole-genome sequencing (WGS) in 142 *P. aeruginosa* clinical isolates recovered from three hospitals in Islamabad and Rawalpindi, Pakistan between 2016 and 2017. Isolates were subjected to antimicrobial susceptibility testing (AST) by Kirby-Bauer disk diffusion, and their genomes were assembled from Illumina sequencing data. $\beta$-lactam resistance was high, with 46% of isolates resistant to piperacillin-tazobactam, 42% to cefepime, 48% to ceftolozane-tazobactam, and 65% to at least one carbapenem. Twenty-two percent of isolates were resistant to all $\beta$-lactams tested. WGS revealed that carbapenem resistance was associated with the acquisition of metallo-$\beta$-lactamases (MBLs) or extended-spectrum $\beta$-lactamases (ESBLs) in the $bla_{GES}$, $bla_{VIM}$, and $bla_{NDM}$ families, and mutations in the porin gene *oprD*. These resistance determinants were found in globally distributed lineages, including ST235 and ST664, as well as multiple novel STs which have been described in a separate investigation. Analysis of AST results revealed that acquisition of MBLs/ESBLs on top of porin mutations had an additive effect on imipenem resistance, suggesting that there is a selective benefit for clinical isolates to encode multiple resistance determinants to the same drugs. The strong association of these resistance determinants with phylogenetic background displays the utility of WGS for monitoring carbapenem resistance in *P. aeruginosa*, while the presence of these determinants throughout the phylogenetic tree shows that knowledge of the local epidemiology is crucial for guiding potential treatment of multidrug-resistant *P. aeruginosa* infections.

**IMPORTANCE** *Pseudomonas aeruginosa* is associated with serious infections, and treatment can be challenging. Because of this, carbapenems and $\beta$-lactam/$\beta$-lactamase inhibitor combinations have become critical tools in treating multidrug-resistant (MDR) *P. aeruginosa* infections, but increasing resistance threatens their efficacy. Here, we used WGS to study the genotypic and phylogenomic patterns of 142 *P. aeruginosa* isolates from the Potohar region of Pakistan. We sequenced both MDR and antimicrobial susceptible isolates and found that while genotypic and phenotypic patterns of antibiotic resistance correlated with phylogenomic background, populations of MDR *P. aeruginosa*

Address correspondence to Saadia Andleeb, saadiamarwat@yahoo.com, Carey-Ann D. Burnham, cburnham@wustl.edu, or Gautam Dantas, dantas@wustl.edu.

The authors declare no conflict of interest.

10.1128/spectrum.00766-22 **1**

were found in all major phylogroups. We also found that isolates possessing multiple resistance mechanisms had significantly higher levels of imipenem resistance compared to the isolates with a single resistance mechanism. This study demonstrates the utility of WGS for monitoring patterns of antibiotic resistance in *P. aeruginosa* and potentially guiding treatment choices based on the local spread of $\beta$-lactamase genes.

**KEYWORDS** *Pseudomonas aeruginosa*, molecular epidemiology, hospital-acquired infections, carbapenem resistance, multidrug-resistant (MDR) *Pseudomonas aeruginosa*, multidrug resistance

The spread of multidrug-resistant (MDR) Gram-negative pathogens is a global public health concern, with carbapenem-resistant *Pseudomonas aeruginosa* specifically deemed a serious threat by the CDC and a Priority 1 pathogen for the development of new antibiotics by the World Health Organization (1, 2). Given the clinical importance of carbapenem-resistant *P. aeruginosa*, a clear understanding of the genomic basis of resistance is critical for epidemiologic purposes. In the United States, most carbapenem resistance in *P. aeruginosa* is due to mutations in the gene for the porin *oprD* coupled to the production of AmpC $\beta$-lactamases (3, 4). In other parts of the world, like the Middle East and the Indian subcontinent, the production of metallo-$\beta$-lactamases (MBLs) and extended-spectrum $\beta$-lactamases (ESBLs) is common (5, 6). Although there has been much focus on carbapenemase-producing Enterobacterales and *Acinetobacter* spp., carbapenemase production is an increasingly appreciated resistance mechanism in *P. aeruginosa* as well (5–10).

Differentiating the two resistance mechanisms (porin mutations versus production of carbapenemases) is important, because of the constant threat posed by the mobilization of carbapenemase genes on self-transmissible plasmids (11–13). Transfer of MBLs on plasmids throughout the environment and into the clinic could be catastrophic and possibly drive an outbreak (14–16). Accordingly, we sought to characterize the genomic determinants of carbapenem resistance in *P. aeruginosa* isolates in an environment where the endemic burden of ARGs and MDR pathogens is high. We then analyzed the population structure, and assayed for phenotypic antibiotic resistance in these isolates, with the primary goal of identifying putative associations between phylogeny and virulence or resistance determinants. As a secondary goal, we sought to associate resistance phenotype to various antibiotics with phylogenomic background and the presence of specific antibiotic resistance genes (ARGs).

## RESULTS

**Sample set.** We collected 350 *P. aeruginosa* clinical isolates from respiratory, urine, soft tissue, blood, and abdominal sources in Pakistan from November 2016 to October 2017 (Table S1). All extensively drug-resistant (XDR; not susceptible to ≥1 antibiotic in all but ≤2 drug classes) and susceptible isolates (~100) and a random selection of 100 MDR isolates were selected for further genomic and phenotypic analysis. After shipment to the US site for characterization, isolates not identified as *P. aeruginosa* by matrix-assisted laser desorption/ionization time-of-flight mass spectrometry (MALDI-TOF MS) were excluded, resulting in 149 isolates for whole-genome sequencing. After sequencing, 7 isolates failed to meet our genome quality cutoffs (Materials and Methods) leaving 142 for analysis. Of these, 85/142 (60%) isolates came from the Pakistan Institute of Medical Sciences (PIMS), 43/142 (30%) came from Railway General Hospital (RGH), and 14/142 (10%) came from Military Hospital Rawalpindi (MH). Most isolates came from medical intensive care units (40/142, 28%), followed by outpatient clinics (33/142, 23%), emergency rooms (25/142, 18%), and medical wards (24/142, 17%) (Table S1). A subset of 6 isolates has been previously analyzed for genomic content (17). However, for internal consistency, the work described here results from *de novo* analyses and phenotyping done with the full sample set.

**Phylogenetic analysis of *P. aeruginosa* isolates.** To determine the relatedness of the collection, we performed whole-genome short-read (Illumina) sequencing on each

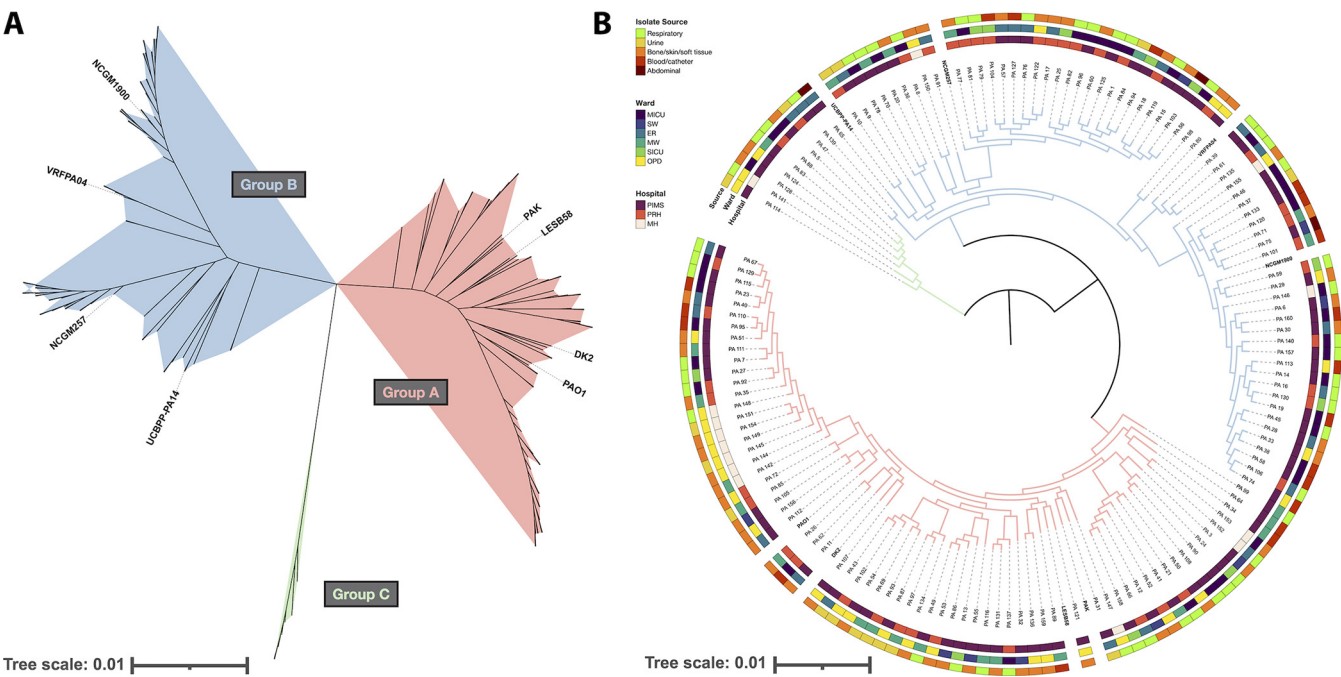

**FIG 1** Phylogenetic analysis of 142 *Pseudomonas aeruginosa* isolates showed segregation into three major clades. (A) Unrooted approximate-maximum likelihood phylogenetic tree constructed based on a core-genome alignment of 142 PA isolates from this study and 8 reference genomes. Tree scale is substitutions per site, and branches are colored by clade: Group A in red, Group B in blue, and Group C in green. (B) Midpoint-rooted circular representation of the same phylogenetic tree, with branch colors representing the same clades as Fig. 1A. Outer rings are annotated with the hospital, ward, and body site each isolate was recovered from. The tree scale indicates substitutions per site.

isolate. After assembly, the draft genomes were used to construct a core genome phylogeny (Fig. 1). Consistent with other studies, we find that most isolates segregate into two large phylogenetic groups: Group A, containing PAO1, and Group B, containing PA14 (Fig. 1A) (18–20). 48% (68/142) of the cohort are in Group A, and 47% (67/142) are in Group B. We also observed a minor clade – Group C – representing only 5% (7/142) of the total collection, which is also consistent with previous analyses of *P. aeruginosa* population structure (19). Multilocus sequence types (STs) were assigned using mlst (21), revealing 41 unique STs among the cohort (Fig. 2A). The most common ST was ST235 (28/142, 20%), followed by ST357 (22/142, 15%), and ST664 (20/142, 14%). Seven new STs were discovered in this cohort (ST3472, ST3489, ST3491, ST3492, ST3494, ST3656, and ST3657) whose allelic profiles have been uploaded to the PubMLST database (22). A separate investigation reported the genomic context surrounding those new STs (17).

To identify putative transmission clusters, we examined the hospital and ward that each isolate was collected from. Some clustering was observed, such as in the six Group A isolates collected from outpatient clinics in MH. However, there was no statistically significant correlation between hospital and population structure (*P* = 0.1391, Fisher's Exact Test) (Fig. 1B). At least one isolate from each group was recovered from each hospital, and closely related isolates were recovered from diverse wards and body sites. For example, PA_106 and PA_74 are closely related (52 pairwise core genome single nucleotide polymorphisms (SNPs), 99.99% ANI), but were recovered from different hospitals, wards, and body sites (Fig. 1B).

**Genotypic and phenotypic β-lactam resistance was found across all phylogenetic groups.** To identify correlations between phylogenetic relatedness and phenotypic antibiotic resistance, we performed antimicrobial susceptibility testing using the Kirby-Bauer disk diffusion method in accordance with the Clinical Laboratory and Standards Institute (CLSI) guidelines on a variety of antibiotics relevant for human and veterinary use (Fig. 2B). To identify a genotypic basis for resistance, we used NCBI AMRFinder to identify acquired ARGs in the draft genomes (Fig. 2B). Overall, the isolates were highly resistant

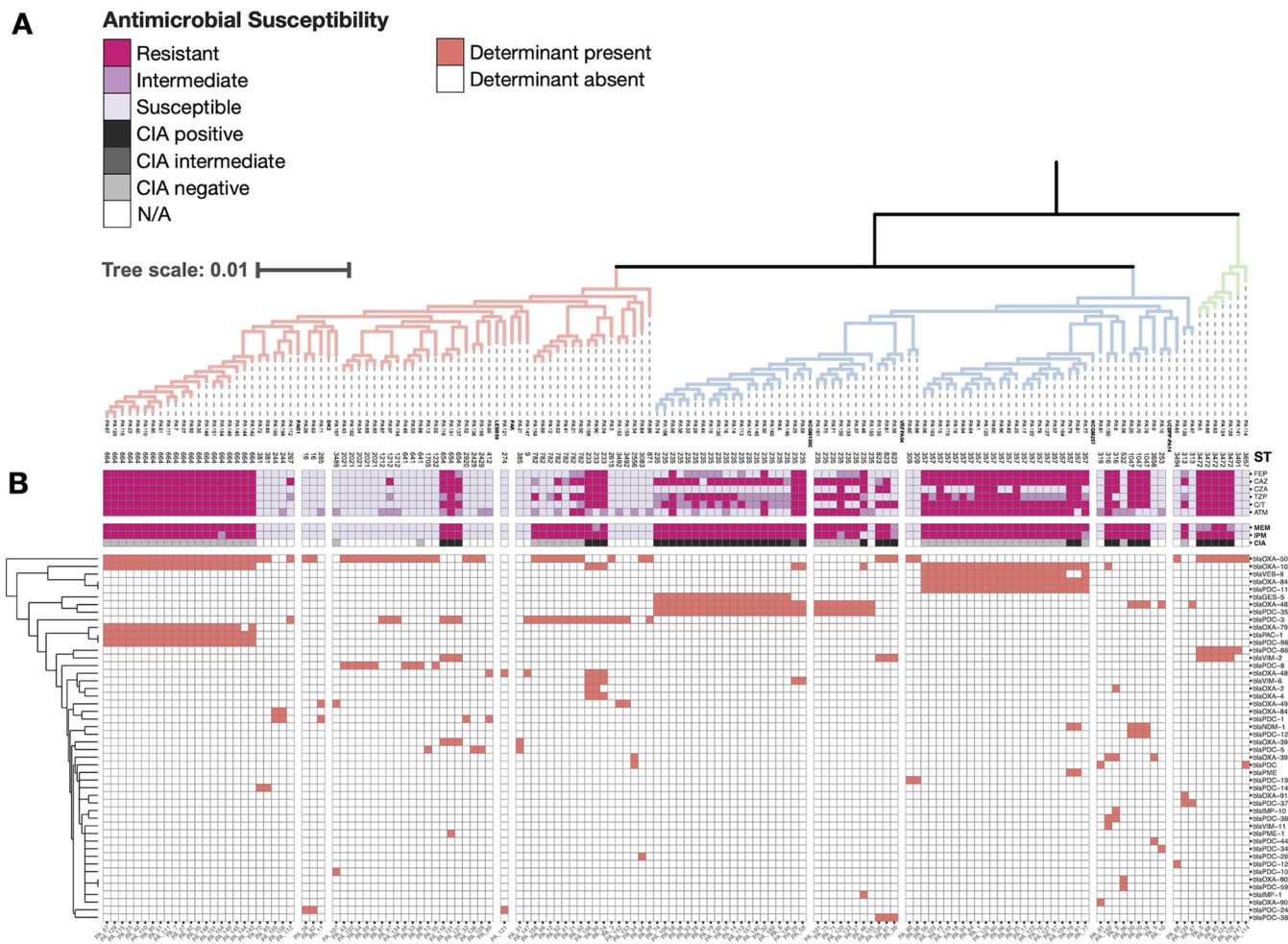

**FIG 2** β-lactam resistance phenotypes and *in silico* resistance determinant identification show a large burden of ARGs. (A) Midpoint-rooted rectangular representation of the same tree in Fig. 1B annotated with MLST group. Branch colors depicting Group A (red), Group B (blue), and Group C (green) have been reproduced. (B) The tree is annotated with the interpretation of AST results (magenta) and the presence/absence matrix of β-lactam ARGs (red), determined by NCBI AMRfinder. ARGs are hierarchically clustered by Euclidean distance, with the dendrogram shown on the left axis. FEP = cefepime, CAZ = ceftazidime, CZA = ceftazidime-avibactam, TZP = piperacillin-tazobactam, C/T = ceftolozane-tazobactam, ATM = aztreonam, MEM = meropenem, IPM = imipenem, CIA = carbapenem inactivation assay.

to β-lactams, including many treatments reserved for serious infections. Among this sub-set of sequenced isolates, resistance was 46% (65/142) to piperacillin-tazobactam, 51% (73/142) to ceftolozane-tazobactam, 34% (49/142) to ceftazidime-avibactam, 63% (90/142) to ceftazidime, 42% (62/142) to cefepime, 42% (60/142) to aztreonam, 65% (92/142) to meropenem, and 65% (92/142) to imipenem. Twenty-two percent (31/142) of isolates were resistant to all β-lactams tested. Notably, these resistance phenotypes tended to cor-relate with phylogeny, with groups of isolates that cluster by core genomes alignment also clustering by antibiotic resistance genotype/phenotype (Fig. 2). But even though these phenotypic clusters tended to correlate with phylogenetic relatedness, groups of re-sistant isolates appeared consistently across different clades. For example, clusters of highly resistant isolates appeared in the Group A clade (PA_67, PA_129, PA_115, etc.), the Group B clade (PA_1, PA_125, PA_60), as well as the Group C clade (PA_83 and PA_124) (Fig. 2A). This could be explained by the diversity of β-lactamase genes found across the sample set. Forty-eight unique β-lactamases were detected across all isolates, which appeared to cluster by phylogeny. The most prevalent β-lactamase was *bla*OXA-50 (found in 64/142 isolates), followed by *bla*OXA-10 (found in 49/142 isolates), and *bla*OXA-488 (found in 33/142 isolates). Included in those 48 unique β-lactamases were 22 *ampC* variants, named *Pseudomonas*-derived cephalosporinases (PDCs), some of which have been found to

confer resistance to antipseudomonal cephalosporins and $\beta$-lactam/$\beta$-lactamase inhibitor combinations (23–25).

Carbapenemase genes were detected in 42/142 isolates. The Class A $\beta$-lactamase $bla_{GES-5}$ was found in 18/142 isolates, and the Class B $\beta$-lactamases $bla_{IMP-1}$ (1/142), $bla_{IMP-10}$ (1/142), $bla_{NDM-1}$ (5/142), $bla_{VIM-2}$ (11/142), $bla_{VIM-6}$ (5/142), and $bla_{VIM-11}$ (1/142) were also detected. There were no isolates with multiple carbapenemase genes detected. Because of the wide array of carbapenemases present in this cohort, identical antibiotic susceptibility profiles were found between isolates that represented very different phylogenetic backgrounds and acquired ARGs.

**_P. aeruginosa_ isolates harbor a mosaic of antibiotic resistance determinants and virulence factors.** In addition to $\beta$-lactamases, many other ARGs were detected in these isolates. The median number of ARGs detected in each isolate was 15, and the median prevalence of each ARG was 4. The isolates PA_8 and PA_46 harbored the most ARGs at 21 (Fig. S1B). In addition to the 48 $\beta$-lactamases, we detected 21 ARGs predicted to have activity against aminoglycosides, 1 against fluoroquinolones, 1 against fosfomycin, 6 against folate-synthesis inhibitors, and 3 against tetracyclines, including the tetracycline destructase _tet_(X7) (26). Because _P. aeruginosa_ is intrinsically resistant to tetracyclines due to its multiple efflux systems (27), the presence of _tet_(A) and _tet_(G) was curious. Upon investigation, those genes were always syntenic with other ARGs, including _floR2_, $bla_{VEB-9}$, and $bla_{VIM-2}$, and mobile genetic elements (MGEs) (Fig. S6). The synteny between _tet_ genes and those other ARGs tended to correlate with phylogeny (Fig. S1). The fact that these isolates have a similar phylogenetic background suggests a potential horizontal gene transfer event in a common ancestor, and the retention of _tet_(A) and _tet_(G) is possibly due to proximity to other ARGs.

We detected 14 additional ARGs predicted to act as multidrug efflux pumps or to have activity against chloramphenicol, rifamycin, bleomycin, or macrolides (Fig. S1B). The fosfomycin resistance gene _fosA_, the aminoglycoside resistance gene _aph(3′)-IIb_, and the multidrug efflux components _mexA_, _mexE_, and _mexX_ were present in 141 of 142 isolates, indicating that they are core genes in this cohort. Coincidentally, the one isolate missing _mexA_ in its genome (PA_147) had a larger zone of clearance toward tetracyclines and trimethoprim-sulfamethoxazole (Fig. S1B). The chloramphenicol acetyltransferase _catB7_ was present in nearly all isolates as well (140/142).

To determine whether the ARG profiles were correlated with the phylogenetic background of the isolates, we used the complete ARG presence/absence table from AMRFinder to calculate the Jaccard distance between each isolate. ARG profiles were significantly different between the 3 clades ($P = 0.001$, PERMANOVA), suggesting that overall ARG content is correlated with the phylogenomic background. To visualize these differences, we performed principal coordinate analysis (PCoA) on the Jaccard distances and found that the first 3 principal coordinates summarized 39.2% of the overall variance (Fig. S2B). Plotting by the first 2 principal coordinates separated the 3 clades (Fig. S2A).

Using the definition of MDR as "not susceptible to $\geq$1 drug in $\geq$3 drug classes" (28), we found that 64% (91/142) of isolates were MDR (when only examining drugs that have CLSI breakpoints). Taken all together, this demonstrates the incredibly high burden of antibiotic-resistant _P. aeruginosa_ isolates in Pakistan that this cohort reflects (6, 29, 30).

To see if specific virulence factors correlated with phylogeny, we identified genes in our cohort that are annotated in the virulence factor database (VFDB) (31). This was not particularly informative because most hits were present in all or nearly all isolates (Table S4 and Fig. S3). However, the presence of some VFs was indicative of population structure, namely, the Type III secretion system (T3SS) effectors _exoS_ and _exoU_ (32, 33). Consistent with a previous study, the presence of _exoS_ was entirely confined to Group A and Group C isolates while _exoU_ was confined entirely to Group B isolates (Fig. S3) (19). While not completely discriminatory, the presence of flagellar components _fliC_, _fliD_, _fliS_, _fliT_, _flgK_, _flgL_, and _flaG_ was associated with Group A isolates (Fig. S3). Those

genes were detected in 74% (50/68) of Group A isolates, 3% (2/67) of Group B isolates, and 0% (0/7) of Group C isolates.

**Carbapenem resistance is associated with porin mutations and acquired carbapenemases.** Carbapenem resistance in *P. aeruginosa* can be conferred both by acquired carbapenemases as well as mutations that inactivate the porin *oprD* (34–37). To elucidate the mechanism of resistance in this cohort, we searched all draft genomes (carbapenem-susceptible and carbapenem-resistant) for single nucleotide polymorphisms (SNPs) and small insertion-deletions (indels) in the *oprD* gene compared to the PAO1 reference genome. All but 3 isolates in this cohort had a non-wild-type *oprD* sequence, with most isolates possessing one or more missense mutations (Fig. S4 and Table S5). Notably, most isolates (83/142, 58%) possessed an inactivating mutation (a frameshift or nonsense variant) in *oprD* (Fig. S4). When we compared these results to the AST and ARG data, we found that a nonsusceptible phenotype to carbapenems was concordant with either an inactivating mutation in *oprD* or the presence of an acquired carbapenemase within this cohort (Fig. 3). Interestingly, a set of isolates (20%, 28/142) had *both* an inactivating *oprD* mutation and an acquired carbapenemase. To explore whether there was a phylogenetic association with specific *oprD* mutations, we extracted the nucleotide sequence of *oprD* from the draft genome, aligned all 142 sequences with the reference PAO1 *oprD* using MAFFT, and constructed a tree with RAxML (38, 39). Unsurprisingly, the *oprD* variants tended to correlate with phylogenetic background, mimicking the clustering seen by core-genome phylogeny (Fig. 3). Interestingly, there were 2 groups of 3 sequences from Group A isolates (PA_64/PA_152/PA_153 and PA_34/PA_99/PA_107) that clustered with sequences from Group B isolates (Fig. 3). Within this cohort, a large majority of the Group B isolates (79%, 53/67) possessed an inactivating *oprD* mutation, while almost half of the Group A isolates (44%, 30/68) and none of the Group C isolates did (Fig. 3). Most of the Group B isolates had either a Met141fs or Trp417fs mutation (Fig. S4).

**A porin mutation alone was associated with high levels of meropenem resistance while the production of a carbapenemase increased imipenem resistance.** We next looked to see if the level of phenotypic resistance to carbapenems in this cohort correlated with the resistance mechanism. When focusing on meropenem, isolates with only an acquired carbapenemase had a median zone of clearance of 15 mm, while isolates with an inactivating *oprD* mutation had a significantly smaller zone of clearance of 6 mm ($P = 6.66e-3$, Dunn's test) (Fig. 4A). Isolates with both a carbapenemase and porin mutation also had significantly smaller zones of clearance compared to isolates with only a carbapenemase ($P = 6.50e-3$, Dunn's test) (Fig. 4A). So, while the presence of a carbapenemase was associated with some phenotypic resistance to meropenem, an inactivating *oprD* mutation was associated with a level of meropenem resistance at the limit of quantification for disk diffusion testing (i.e., 6 mm, the diameter of the disk).

When focusing on imipenem, isolates with a carbapenemase alone had a median zone of clearance of 11 mm, and isolates with a porin mutation alone had a similar zone at 10 mm (Fig. 4A). However, isolates that had both a carbapenemase and inactivating porin mutation had a significantly smaller zone at 6 mm, compared to isolates with a carbapenemase or porin mutation alone ($P = 6.98e-3$ and $P = 2.67e-3$, respectively, Dunn's test). This indicated that for this cohort, each mechanism had an additive effect on imipenem resistance, where isolates that possessed both resistance mechanisms had significantly smaller zones of clearance than each mechanism alone (Fig. 4A and B).

To determine if this effect was limited to any particularly prevalent ARGs, we subsetted the data by carbapenemase presence. The presence of $bla_{IMP-10}$ and $bla_{VIM-11}$ alone was associated with levels of resistance toward both imipenem and meropenem that were at the lower limit of quantification (Fig. 4C). However, the additive effect of the porin mutation in resistance is apparent for isolates with $bla_{VIM-2}$ and $bla_{VIM-6}$ (Fig. 4C). Interestingly, one isolate with $bla_{IMP-1}$ (PA_46) was resistant to meropenem but susceptible to imipenem (Fig. 2B). It should be noted, though, that all isolates with

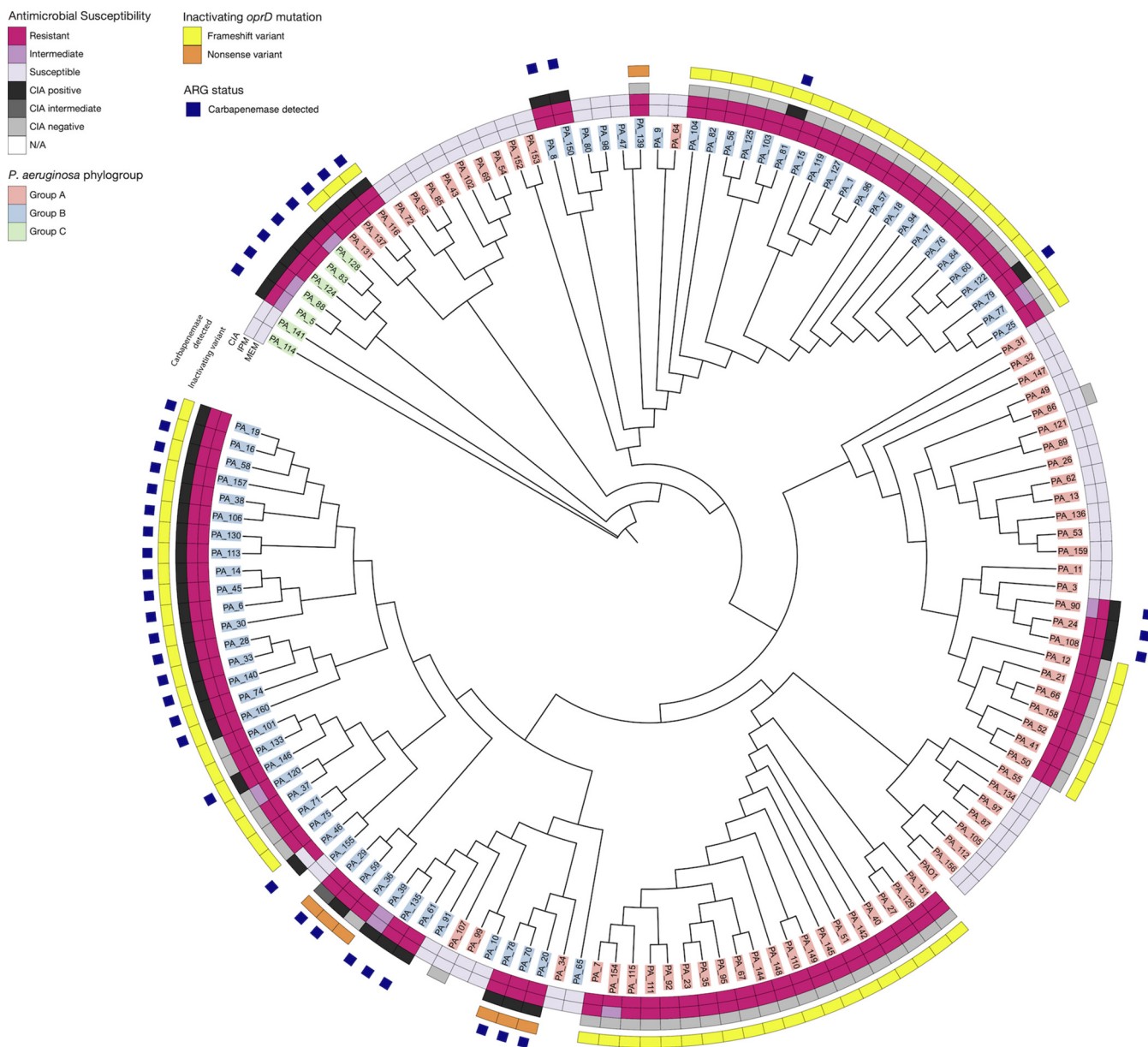

**FIG 3** Inactivating mutations in *oprD* were associated with carbapenem resistance phenotype. Circular cladogram constructed from a multiple sequence alignment of all *oprD* variants observed in this cohort. Leaves are colored by the phylogroup of the isolate where the gene came from. AST results are annotated on inner rings. Variants that are inactivating (frameshift or nonsense mutant) are indicated as yellow or orange boxes. The outermost ring indicates whether a carbapenemase ($bla_{GES-5}$, $bla_{IMP-1}$, $bla_{IMP-10}$, $bla_{NDM-1}$, $bla_{VIM-2}$, $bla_{VIM-6}$, or $bla_{VIM-11}$) was detected by NCBI AMRfinder as a blue box. Altogether, these annotations highlight that all carbapenem-resistant phenotypes are associated with either an inactivating *oprD* mutation, the presence of a carbapenemase, or both. For this cohort, CIA results are in complete concordance with the genotypic presence of a carbapenemase.

inactivating *oprD* mutations or a carbapenemase were not susceptible to both carbapenems by CLSI breakpoints, except for PA_46.

**Mutations in efflux pump regulators did not affect carbapenem resistance when controlling for *oprD* mutations.** In addition to import porins, regulation of efflux pumps is an important resistance mechanism. In *P. aeruginosa*, overexpression of multidrug efflux pumps, specifically the tripartite efflux systems MexAB-OprM, MexXY-OprM, and MexCD-OprJ, are associated with carbapenem resistance (27, 40, 41). Of these systems, MexAB-OprM is the most important for carbapenem resistance, and mutations in the repressors *mexR*, *nalC*, and *nalD* (41, 42). There are a plethora of mutations associated with increased MexAB-OprM expression, but strong phenotypes are associated with frame-shifting indels and early stop codons which derepress the

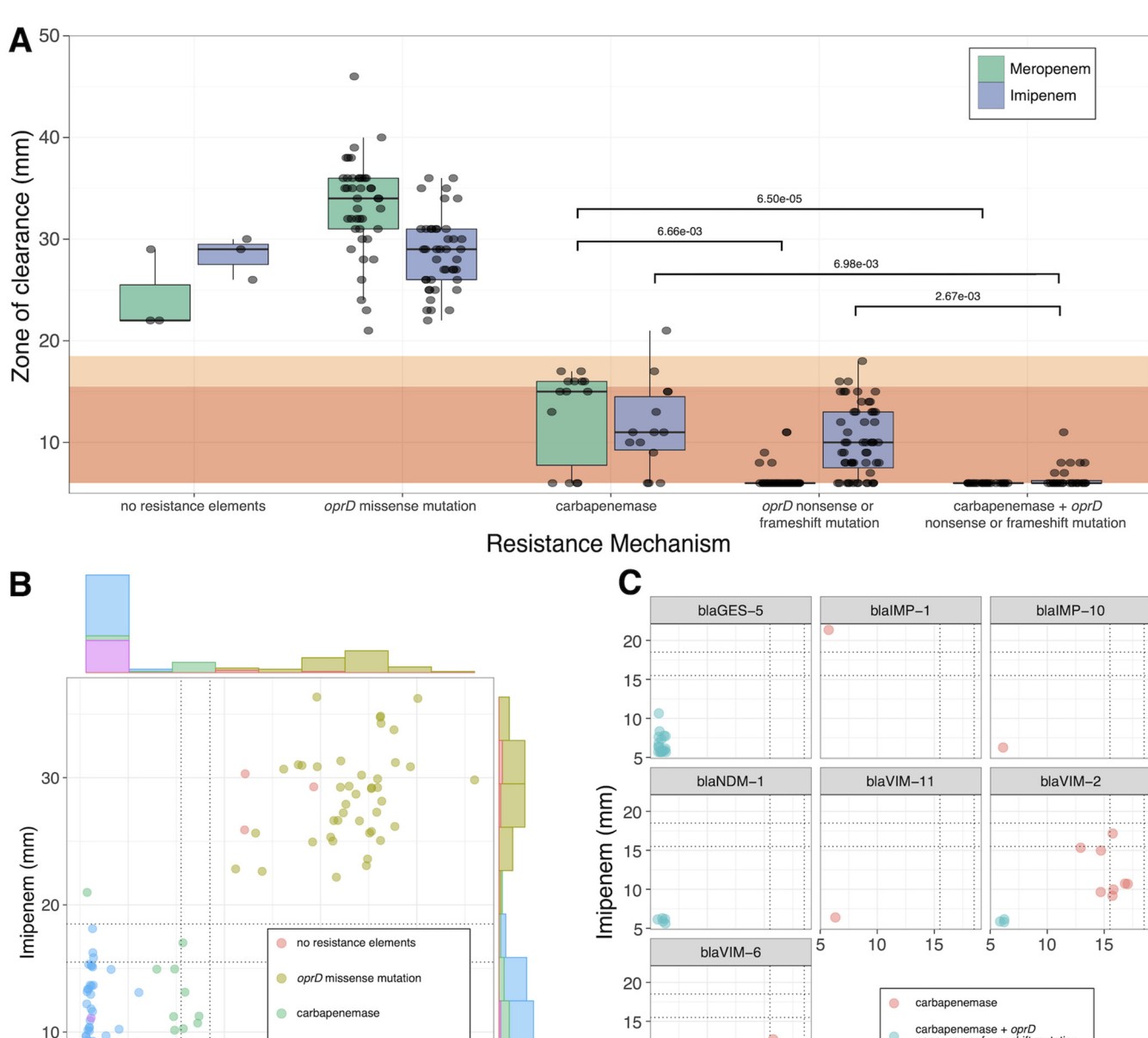

**FIG 4** Presence of an inactivating mutation in oprD is associated with meropenem resistance, while carbapenemase expression has an additive effect on imipenem resistance. (A) Box and whisker plot comparing zone of clearance against carbapenems for isolates with different resistance mechanisms. A small amount of jitter added to points for readability. Background colors correspond to resistant (dark orange) and intermediate (light orange) susceptibility, per CLSI guidelines. For meropenem, the presence of an inactivating *oprD* mutation alone is associated with a smaller zone of clearance compared to the production of a carbapenemase alone ($P = 6.66e{-}3$, Dunn's test). In contrast, the production of a carbapenemase has an additive effect on imipenem resistance: isolates with only an inactivation *oprD* mutation or only carbapenemase production have similar zones of clearance (NS, Dunn's test). However, isolates that possess both resistance mechanisms display significantly smaller zones of clearance compared to a single mechanism. (B) XY scatterplot showing the same data from Fig. 4A, with marginal histograms added to each axis. The small amount of jitter added to points for readability. Dotted lines are drawn at CLSI breakpoints for "resistant" and "intermediate". The zone of clearance for the imipenem varies much more than meropenem for isolates with an *oprD* mutation. (C) XY scatterplot, subsetted by carbapenemase. A small amount of jitter added to points for readability. The presence of *bla*$_{VIM-11}$ and *bla*$_{IMP-10}$ alone was associated with zones of clearance at the limit of detection. For carbapenems that were observed in *oprD*+ and *oprD*− backgrounds, isolates with both resistance mechanisms displayed smaller zones of clearance for both drugs.

expression of these efflux systems (43–45). To determine if these mutations were present in this cohort, we searched the draft genomes for SNPs and indels in the *mexR*, *nalC*, and *nalD* genes compared to the PAO1 reference genome. Most isolates in this cohort had some *mexR* mutation (100/142, 70%), and nearly all had a *nalC* mutation

(139/142, 98%). We observed 3 different missense mutations in *mexR*, the most common being V126E (present in 99/142 isolates, 70%), which has been not found to be correlated with increased *mexA* expression or antibiotic resistance (Table S6) (41, 46–48). The other two variants (G101E and A103G) were only observed in one and seven isolates, respectively, and both were observed in isolates that were susceptible to imipenem and meropenem (PA_156 and PA_114). This suggests that *mexR* mutations were unlikely to significantly influence carbapenem susceptibility in this cohort. When examining *nalC*, we observed seven different missense mutations and a frameshift mutation (Table S7), but the only isolate possessing that variant (PA_99) was susceptible to both carbapenems tested (Fig. 2), suggesting that inactivation of *nalC* was not relevant to carbapenem resistance in this cohort. There were more unique variants in *nalD* (16, Table S8), which were observed in about half the genomes in this cohort (74/142, 52%). Because many of these were frame-shifting deletions (most commonly 398_399del), we examined their effect on carbapenem zone sizes in this cohort, compared to inactivating *oprD* mutations and the presence of carbapenemases. When examining only isolates without carbapenemase genes in their genome, we found no significant difference in the zone of clearance between isolates with a wt or missense *nalD* mutation and isolates with an inactivating mutation in *nalD*, when controlling for the presence of an inactivating *oprD* mutation (Fig. S5A) ($P > 0.05$, Dunn's test). Put another way, the difference in the zone of clearance for isolates with wt *nalD* and isolates with an inactivating *nalD* mutation was only significant if one of those isolates also had an inactivating *oprD* mutation. This was true for both meropenem and imipenem (Fig. S5A).

Although MexAB-OprM is the most important efflux pump for carbapenem resistance, we also considered mutations that increase the expression of other pumps. Mutations in the repressors *mexZ* and *nfxB* are associated with increased expression of MexXY-OprF and MexCD-OprJ, respectively (27, 41). We detected *nfxB* variants in only 12 genomes, only one of which possessed a frameshift variant that was susceptible to meropenem and imipenem (PA_105, Table S9). We detected *mexZ* mutations in 83 genomes, including numerous inactivating mutations. Those mutations included six unique frameshift variants and a 21-bp deletion resulting in a gene fusion between *mexZ* and the adjacent gene (PA2021) (Table S10). We repeated the earlier analysis that we performed on *nalD*, and similarly found no significant difference in the zone of clearance between isolates with a wt or missense *mexZ* mutation and isolates with an inactivating mutation in *mexZ* (Fig. S5B). This suggests that within the limit of detection of this study, *oprD* mutations are much more consequential for meropenem and imipenem resistance than mutations in efflux pump regulators.

## DISCUSSION

In many parts of the world, including the United States, carbapenem resistance in *P. aeruginosa* can largely be attributed to mutations in porins, overproduction of AmpC $\beta$-lactamases, and multidrug efflux pumps (3, 4). In Pakistan, where the endemic burden of MBLs and other carbapenemases is much higher, carbapenem resistance may be associated with a greater variety of mechanisms (49–51). This necessitates more thorough molecular epidemiological investigations to characterize and combat this threat, as these mechanisms have already spread outside the Indian subcontinent (9, 14, 52–55).

In this study, we sought to characterize the genomic determinants of carbapenem resistance in *P. aeruginosa* in an environment where the endemic burden of ARGs and MDR pathogens is very high, and where we had previously observed diverse mechanisms of carbapenem resistance (5, 6). Although this is not the first study to focus on carbapenem resistance in *P. aeruginosa* in Pakistan (49–51), it is to our knowledge the most comprehensive comparative genomic analysis paired with antibiotic susceptibility results.

We observed that in our cohort, the ARG profiles correlated with phylogeny (Fig. S2).

This is consistent with other independent observations that link specific ARGs to *P. aeruginosa* population structure (56). For example, the large group of ST664 isolates in this study all possessed the same inactivating *oprD* mutation, $bla_{PDC-98}$, $bla_{OXA-10}$, and $bla_{PAC-1}$, and demonstrated resistance to all $\beta$-lactams tested (Fig. 2 and Fig. S3). However, a combination of porin mutation and acquired $\beta$-lactamase was not limited to ST664 but was found in all 3 clades, and importantly, carbapenemase-producing *P. aeruginosa* was found in all 3 clades. This suggests that there is no inherent phylogenomic barrier to, or correlation with, acquisition of high-risk carbapenemases. The consequences of this acquisition can be demonstrated by the set of ST235 isolates in this cohort, which possessed multiple resistance mechanisms. All ST235 isolates in this cohort possessed an *oprD*-inactivating mutation, except for one (PA_155), which is associated with resistance to meropenem and imipenem, but isolates lacking a carbapenemase remained susceptible to ceftolozane-tazobactam (Fig. 2B). Acquisition of $bla_{GES-5}$ was associated with resistance to ceftolozane-tazobactam, and acquisition of $bla_{VIM-6}$ or $bla_{IMP-1}$ was associated with nonsusceptibility to all other $\beta$-lactams (Fig. 2).

From a clinical perspective, this presents a challenge as it becomes harder to differentiate "carbapenem-resistant" from "carbapenemase-producing" *P. aeruginosa* based solely on phenotype. Guidelines for predicting carbapenemase-producing *P. aeruginosa* from AST profiles have been suggested, but this becomes more difficult as resistance mechanisms become more diverse (57, 58). Combinations of porin mutations, various acquired carbapenemases, and $bla_{PDC}$ variants in *P. aeruginosa* can result in isolates with resistance to carbapenems, cephalosporins, penicillins, and $\beta$-lactam/$\beta$-lactamase inhibitor combinations. Different combinations of these mechanisms are not only possible but prevalent in this geography. This highlights the importance of thorough susceptibility testing (including carbapenem inactivation assays), and genomic surveillance to monitor and combat the spread of these high-risk ARGs and organisms. Notably, ST664 isolates possessing $bla_{PAC-1}$ and demonstrating high levels of $\beta$-lactam resistance have been reported as originating from Afghanistan and Mauritius, suggesting that this is not limited to Pakistan (59). Knowledge of the genotypes that are prevalent in a geographic location is important for evaluating the use of various $\beta$-lactam therapies in that setting.

The consequences of carbapenemase production were highlighted in our observation that acquiring multiple resistance mechanisms was associated with higher levels of imipenem resistance than the individual mechanisms alone (Fig. 4). Although *oprD* mutations are well studied in *P. aeruginosa*, to our knowledge this is the first comparison between the level of resistance in isolates with different and combinatorial resistance mechanisms. While interesting, this behavior appears to be most notable in the carbapenemases $bla_{VIM-2}$ and $bla_{VIM-6}$, while carbapenemases like $bla_{VIM-11}$ and $bla_{IMP-10}$ are associated with small zones of clearance on their own (Fig. 4C). These results would be consistent with the observation that $bla_{VIM-11}$ has higher catalytic efficiency than $bla_{VIM-2}$ toward carbapenems (60). This further highlights the importance of genomic surveillance because even genetically similar ARGs may affect clinically relevant bacterial fitness differently. Further work with isogenic strains could better disentangle the complex interaction between carbapenemase, porin, efflux pump expression, and resistance phenotypes.

One limitation of this study was that because it was a cross-sectional, observational study, we were unable to show the effects of individual ARGs in a common genetic background. Differences are inferred from the cohort and are subject to biases associated with the specific phylogenomic background of the sequenced isolates. Additionally, because we only used short-read Illumina sequencing, we were unable to unequivocally implicate ARGs as present on mobilizable plasmids, which often require long scaffolding reads for complete assembly across repetitive elements (61–63). Further, because we sequenced exclusively clinical isolates in this study, we are unable to discuss the burden of these ARGs in related built or natural environments. Although, our groups have previously shown that MDR *P. aeruginosa*, as well as other bacteria involved in nosocomial infections, can persistently colonize hospital surfaces

in this geographic area (6). Finally, because we used disk diffusion to assess antimicrobial susceptibility instead of quantitative broth microdilution, we are not able to report MIC values.

In this study, we have demonstrated the utility of WGS for monitoring patterns of antibiotic resistance in *P. aeruginosa*, and the complex interplay of efflux pumps, porins, and acquired carbapenemase genes on resistance. By combining the genomic data with antibiotic susceptibility data, we identified a unique relationship between combinations of genomic markers and phenotypic resistance. The key findings of this study are (i) overall antibiotic resistance determinants are associated with a phylogenetic background in *P. aeruginosa*, (ii) although carbapenem resistance in *P. aeruginosa* is associated with specific genomic markers, they are found across the phylogenetic tree, and (iii) combinations of inactivating porin mutations and acquired carbapenemases are associated with smaller zones of clearances toward imipenem. These findings further motivate the tracking of genomic and phenotypic features of MDR *P. aeruginosa* across the globe, both to guide the treatment of local strains and assess potential outbreak risks. Further work with prospective tracking of clinical and environmental isolates is necessary to determine the extent to which these resistance determinants can spread throughout the local geography.

## MATERIALS AND METHODS

**Sample collection and culturing.** A total of 350 archived *P. aeruginosa* isolates were collected from pathology labs from three different hospitals in Islamabad and Rawalpindi, Pakistan: Pakistan Institute of Medical Sciences (PIMS), Railway General Hospital (RGH), and Military Hospital Rawalpindi (MH). Dilute suspensions of bacterial culture were prepared using normal saline, and 100 $\mu$L was plated on both Blood Agar (Oxoid) and MacConkey's Agar (Oxoid, UK). The plates were incubated at 37°C for 24 h. Morphologically distinct colonies were substreaked on *Pseudomonas* cetrimide agar (Oxoid) and incubated for 24 h at 37°C. Isolates were classified as antibiotic susceptible, extensively drug-resistant (XDR, not susceptible to ≥1 antibiotic in all but ≤2 drug classes), or MDR (not susceptible to ≥1 antibiotic in ≥3 more drug classes) (28). Drug classes used to define XDR and MDR are defined in Table S2. All XDR and susceptible isolates (~100) and a random selection of 100 MDR isolates were selected for further genomic and phenotypic analysis. After shipment to the US site for characterization, species identity and isolate purity was confirmed by plating to sheep's blood agar and performing identification using matrix-assisted laser desorption/ionization time-of-flight mass spectrometry (MALDI-TOF MS) with Vitek MS v2.3.3 system (bioMérieux, Durham, NC, USA). Isolates not identified as *P. aeruginosa* were excluded, resulting in 149 isolates for whole-genome sequencing.

**Reference genomes.** To strengthen the rooting and overall structure of the phylogenetic tree, eight reference genomes were selected from the collection described by Subedi et al. (18). Four genomes were selected from each of the two major phylogroups described in that study to represent a global pangenome. To ensure only high-quality reference genomes were used, we selected assemblies marked as "complete" or "gapless chromosome" in the *Pseudomonas* Genome Database (64).

**Antibiotic susceptibility testing.** After shipment to the US site, antimicrobial susceptibility testing was performed using Kirby-Bauer Disk Diffusion, interpreted according to criteria from the M100-S30 (65).

**Illumina whole-genome sequencing.** Total genomic DNA was obtained from pure *P. aeruginosa* cultures using the QIAmp BiOstic Bacteremia DNA kit (Qiagen, Germantown, MD, USA). DNA was quantified with the Quant-iT PicoGreen dsDNA assay (Thermo Fisher Scientific, Waltham, MA, USA), and 0.5 ng of genomic DNA was used to create sequencing libraries with the Nextera kit (Illumina, San Diego, CA, USA) using a modified protocol (66). Samples were pooled and sequenced on the Illumina NextSeq platform to obtain 2 × 150 bp reads. The reads were demultiplexed by barcode and had adapters removed with Trimmomatic v0.38 (67). Processed reads were *de novo* assembled into draft genomes with Unicycler v0.4.7 (67) using default settings and the assembly.fasta file was used for all downstream analysis. Assembly quality was verified using QUAST v4.5 (68), and CheckM v1.0.13 (69). Genomes were included for analysis if the assemblies (i) had an average coverage (read depth) >20×, (ii) had a total length within 20% of the *P. aeruginosa* PAO1 genome size (5 to 7.56 Mbp), (iii) the length of the assembly in small (<1 kbp) contigs represented <2% of the total assembly length, and (iv) estimated completeness >95% and contamination <5%.

***In silico* analysis.** Protein-coding sequences were predicted with prokka v1.14.5 (70). Core genome alignment was generated using Roary v3.12.0 (71). Coding sequences were clustered at 95% identity, and the 4,509 core genes were used to create a core genome alignment with PRANK v1.0 (72). To ensure that the tree was appropriately rooted, the reference genomes selected were from isolates that span the *P. aeruginosa* pangenome, including the commonly studied PAO1, PA14, and LESB58 (73–78) (Table S3). The core genome alignment was converted to an approximate maximum-likelihood tree using FastTree v2.1.10 (79) and viewed in iToL (80). Multilocus sequence types (STs) were assigned using PubMLST typing schemes with mlst v2.19 (21, 22). Acquired antibiotic resistance genes were identified using AMRFinderPlus v3.9.8 (81). Annotation of MGEs on contigs with *tet* genes was done by querying the prokka annotation for mention of the following terms: "transposase," "transposon," "integrase,"

"integron," "conjugative," "conjugal," "recombinase," "recombination," "mobilization," "phage," "plasmid," "resolvase," "insertion element," and "mob". If a prokka annotation was not available, the closest Pfam annotation was used. Virulence genes were predicted by querying the predicted protein sequences against the virulence factor database (VFDB) for *Pseudomonas* spp. with 90% identity and 90% coverage cutoffs (31). OprD mutations were identified by mapping reads from clinical isolates to *P. aeruginosa* PAO1 reference genome (73) using snippy v4.6.0 (82) and filtering for locus tag PA0958. Nucleotide sequences for *oprD* were identified in draft genomes by blastn using the PAO1 *oprD* sequence as query and extracted from assemblies using BEDtools (83). Sequences were hand curated to ensure the entire length of the gene was represented in hit, including those that were frame-shifted. Multiple sequence alignment was performed on extracted sequences using MAFFT, and a maximum likelihood tree was constructed a tree using RAxML (38, 39).

**Statistical analyses.** All statistical tests were performed using the *stats*, *vegan*, and *FSA* packages in R (84–86). Comparisons of zones of clearance were performed by grouping the AST data based on the presence of mutations in the genes of interest (*oprD*, *nalD*, etc.) in the host genome and performing a Kruskal-Wallis test. If that comparison was significant ($P < 0.05$), it was followed by Dunn's test of multiple comparisons with Benjamini-Hochberg adjustment, and the adjusted $P$ value was used to determine the significance of the individual comparisons.

**Data availability.** Assemblies and sequencing reads associated with this report are available from NCBI under BioProject accession code PRJNA800087. The 6 genomes that were previously reported resulted from an independent *de novo* assembly and are, therefore, available as independent accession IDs (17).

## SUPPLEMENTAL MATERIAL

Supplemental material is available online only.
**SUPPLEMENTAL FILE 1**, PDF file, 0.5 MB.
**SUPPLEMENTAL FILE 2**, PDF file, 0.2 MB.
**SUPPLEMENTAL FILE 3**, PDF file, 0.2 MB.
**SUPPLEMENTAL FILE 4**, PDF file, 0.2 MB.
**SUPPLEMENTAL FILE 5**, PDF file, 0.7 MB.
**SUPPLEMENTAL FILE 6**, PDF file, 0.1 MB.
**SUPPLEMENTAL FILE 7**, XLSX file, 0.3 MB.

## ACKNOWLEDGMENTS

This work was supported by a United States Agency for International Development award (award number 3220-29047) to S.A., C.A.B., and G.D. This work was also supported in part by awards to G.D. through the National Institute of Allergy and Infectious of the National Institutes of Health (NIH) under award numbers U01AI123394 and R01AI155893. L.D.T. received support from F30AI157161. The content is solely the responsibility of the authors and does not necessarily represent the official views of the funding agencies.

We thank The Edison Family Center for Genome Sciences & Systems Biology staff, Eric Martin, Brian Koebbe, Jessica Hoisington-López, and MariaLynn Crosby for their technical support in high-throughput computing and sequencing expertise. We also thank the pathology lab staff and technicians of PIMS, PRH, and MH for providing clinical isolates and assistance with collection and culturing. We thank members of the Dantas lab for their helpful comments and discussion of the manuscript.

We declare no conflict of interest.

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
