## [Reviewer comments · Microbiology Spectrum]

Microbiology Spectrum

Genomic surveillance of clinical *Pseudomonas aeruginosa* isolates reveals an additive effect of carbapenemase production on carbapenem resistance.

Luke Diorio-Toth, Sidra Irum, Robert Potter, Meghan Wallace, Muhammad Arslan, Tehmina Munir, Saadia Andleeb, Carey-Ann Burnham, and Gautam Dantas

Corresponding Author(s): Luke Diorio-Toth, Washington University in St. Louis, School of Medicine

Review Timeline:

Submission Date:

April 14, 2022

Accepted:

May 1, 2022

Editor: Daria Van Tyne

Reviewer(s): The reviewers have opted to remain anonymous.

Transaction Report:

DOI: <https://doi.org/10.1128/spectrum.00766-22>

May 1, 2022

Dr. Luke Diorio-Toth
Washington University in St. Louis, School of Medicine
Edison Family Center for Genome Sciences and Systems Biology
4515 McKinley Avenue
5th Floor, Room 5121
St. Louis, MO 63110

Re: Spectrum00766-22 (Genomic surveillance of clinical *Pseudomonas aeruginosa* isolates reveals an additive effect of carbapenemase production on carbapenem resistance.)

Dear Dr. Luke Diorio-Toth:

Your manuscript has been accepted, and I am forwarding it to the ASM Journals Department for publication. You will be notified when your proofs are ready to be viewed.

Sincerely,

Daria Van Tyne
Editor, Microbiology Spectrum

Journals Department
Supplemental Tables: Accept
Supplemental Figure S2: Accept
Supplemental Figure S1: Accept
Supplemental Figure S6: Accept
Supplemental Figure S3: Accept
Supplemental Figure S4: Accept
Supplemental Figure S5: Accept